# Goat Production, Supply Chains, Challenges, and Opportunities for Development in Vietnam: A Review

**DOI:** 10.3390/ani13152546

**Published:** 2023-08-07

**Authors:** Viet Don Nguyen, Cong Oanh Nguyen, Thi Minh Long Chau, Dinh Quang Duy Nguyen, Anh Tuan Han, Thi Thanh Huyen Le

**Affiliations:** 1School of Environmental and Rural Science, The University of New England, Armidale, NSW 2351, Australia; 2Faculty of Animal Science, Vietnam National University of Agriculture, Hanoi 12406, Vietnam; ncoanh@gmail.com; 3Agricultural Systems Division, Western Highlands Agriculture and Forestry Science Institute, Dak Lak 63124, Vietnam; longchau76@yahoo.com; 4National Centre for Marine Breeding in Central Vietnam, Research Institute for Aquaculture No. 3, Khanh Hoa 57110, Vietnam; haisamduy@yahoo.com; 5Department of Livestock System and Environment Research, National Institute of Animal Science, Hanoi 11913, Vietnam; tuanhavcn@gmail.com (A.T.H.); lehuyen1973@yahoo.com (T.T.H.L.)

**Keywords:** demand, goat production, marketing, smallholder, stakeholders, supply chain

## Abstract

**Simple Summary:**

Over the last decade, the total goat population in Vietnam increased more than two-fold, from 1.29 million to 2.65 million heads. In spite of a transformation from extensive grazing to cut-and-carry intensive systems, more than three-fourths of goats in Vietnam are raised by small-scale producers. The demands for goat meat and milk are significantly increasing, leading to pressure for imports. Goat marketing primarily takes place through informal channels and is dominated by small-scale producers and traders. The formal goat market is poorly developed. The marketing research and statistical data on the goat value chain are scarce. This situation has led to an inconsistency in livestock supply and quality and, as a result, unstable pricing. Information and documentation regarding both horizontal and vertical linkages in the supply chain remain limited. Despite receiving strong support from the government and experiencing high demand, goat production and marketing systems have yet to achieve their full potential. This review overviews the current status of goat production and supply chains in Vietnam. It also identifies the main challenges and opportunities of, and provides suggestions through which improve, the nation’s goat production and marketing.

**Abstract:**

The current situation of goat production and supply chains in Vietnam, along with its difficulties and possibilities, is presented in this review paper. The data and reports of government agencies, scientific journals, and websites were analysed in order to determine the prevailing situation in goat production and marketing. Goats are mainly raised on small-scale farms (73.4% of the total goat population). Goat production is transforming from extensive grazing to cut-and-carry intensive systems. Goat meat and milk supplies have not fully met domestic demand. However, the scale of the domestic market is difficult to ascertain, due to the lack of market research and statistics. Goat marketing is mostly informal and overwhelmingly conducted by small-scale producers and traders, although there are numerous governmental agencies at both the national and local levels regulating formal marketing. The major challenges facing the goat industry are feed shortage; supply inconsistency; limited market infrastructure and research; a lack of sustainable breeding programmes, price incentives, and processing facilities; and competition from foreign suppliers. However, there are opportunities to expand and develop the industry, such as consumers’ health consciousness, increasing demand, high-value adding, and strong government support.

## 1. Introduction

In Vietnam, about 27.6% of the national labour force, equivalent to 14 million people, participated in the agricultural sector in 2021 [1], which accounted for around 12.4% of the gross domestic product (GDP). Within the agricultural sector, about one-quarter was attributed to the livestock sector, which can be associated with the livelihoods of nearly 10 million households across the country [2]. The main domestic animals raised in Vietnam were pigs, poultry, and cattle [3].

Over the past five decades, global goat populations have witnessed a substantial growth of approximately 240%, whereas other livestock species populations have either remained stable or experienced a decline. Presently, the worldwide goat population exceeds 1 billion, with a significant majority, of over 94%, located in Asia (556 million heads) and Africa (388 million heads) [4]. Monteiro et al. (2017) [5] stated that approximately 97% of goat meat was produced by developing countries, led by China with about 38% of the global goat production. In 2019, the total goat population in Southeast Asia was about 39 million heads, led by Indonesia with nearly 19 million goats [6]. Vietnam currently has approximately 2.65 million goats [7], while its neighbouring countries have small national goat populations (640 thousand heads in Laos and 26 thousand heads in Cambodia) [8,9]. These studies agreed that goat industries have considerably expanded and will continue to do so in the coming decade. 

In Vietnam, the goat population is estimated to have increased more than two-fold over the last decade. As a low-input industry, goat production has played a vital role in food security, hunger eradication, and poverty alleviation, generating income for thousands of rural households [10,11]. Furthermore, goats can thrive in harsh environments and can eat a wide range of forages. As a result, most goats were raised in hilly and mountainous areas [12,13], where biodiversity and poverty levels are high [14,15]. Goats in Vietnam depend mainly on growing grasses, crop residues, and the natural browsing of shrubs and bushes. The majority of them are raised on smallholder farms, with fewer than 10 heads/household [16,17]. The main indigenous goat breeds are Co and Bach Thao. Since the 1980s, Boer, among other exotic goat breeds, have been imported and have served as an established breed for intensive meat and milk production. In the past, extensive communal land grazing was the main goat-raising method. Recently, there has been a transformation in the production scale and the farming system [13,18]. Goat products are not essential ingredients in day-to-day diets in the country. They are mainly consumed in restaurants and at special social events [19].

The statistical data on goat production and marketing are limited [20]. Most statistics on goats are derived from formal market data, provided by the Department of Livestock Production and the General Statistics Office (GSO). The bulk of goat markets are domestic and are driven by domestic demand. As domestic supplies are nowhere near sufficient, goats and goat products are imported from Laos, Mongolia, and Australia [21]. There is a lack of marketing research and not enough comprehensive data pertaining to the various participants in the market chain, transaction expenses, and specific details regarding the movement of goats and goat products. The present article aimed to provide an overview of goat production and supply chains, as well as to determine the main challenges and opportunities for their development in Vietnam. Furthermore, suggestions for enhancing goat production and marketing in Vietnam are also provided.

## 2. Methodology

Annual statistical data and production reports were collected from the General Statistics Office of Vietnam, the Provincial Departments of Agriculture and Rural Development, and the Department of Livestock Production. Related marketing reports and scientific publications were also obtained from prestigious journals and websites. From the analyses of data and the collation of various reports, the results were visualised using graphical techniques. Then, an inductive thematic analysis method, described by Braun and Clarke [22], was employed in order to identify, analyse, and discuss common themes, opportunities, and challenges that occurred repeatedly in the previous published reports and articles on goat production and supply chains in Vietnam. 

## 3. Goat Production

### 3.1. Goat Population and Distribution

In Vietnam, there is a tendency toward an increase in goat production. The total goat population increased, from 1.29 million heads to 2.65 million heads, between 2010 and 2020 [7], equivalent to an average annual increase of 10.5% (Figure 1). The highest growth rate was observed during the period from 2015 to 2017. Although dairy goats represent only 2–5% of the national goat population, this sector is also estimated to have experienced a comparable rate of increase in population size [23], from 65 thousand goats in 2010 to 133 thousand goats in 2020. This demonstrates that goat production has received significant attention in Vietnam in recent years. However, the growth level has not met the aspiration of the Vietnamese government of reaching 3.7 million goats in 2018 [24]. The Department of Livestock Production [25] reported that no other livestock species experienced such a growth rate during the same time period. The annual growth of the national poultry herd was 6.5%, from 300.5 million heads to 496.0 million heads. The cattle population slightly increased, from 5.8 million heads to 6.1 million heads, while the sizes of the pig and buffalo populations decreased. These trends could be caused by reasons such as unstable output prices, competition from imported products, serious disease outbreaks, and government policies [26,27,28]. A number of studies explained that increasing production costs and environmental pollution, as well as a lack of capital, land, high-quality feed sources, productive breeds, and labour, are the other major constraints of high-input and concentrate-fed livestock production, such as those of pigs, poultry, and cattle [29,30,31].

Goats are raised mainly in hilly and mountainous regions. According to the GSO (2022) [32], the country’s goat population was raised mainly in the Northern Mountains in 2021, an area representing 734.5 thousand heads, which accounts for 27.5% of the total herd; followed by the Central Coast, with 667.6 thousand heads, which accounts for a quarter of the total herd. The total herd of these two regions combined accounted for 52.5% of the total national goat herd. These two regions have large areas of land comprising forests, mountains, and hills [14], which are suitable for raising goats, and people there have traditionally raised goats. Currently, the goat herd in the southern regions tends to increase in line with the market demand for goat meat. The goat populations in the Southeast region and the Mekong River Delta accounted for 17.7% and 16.2% of the total national herd, respectively. The Red River Delta had the lowest goat population with only 98.1 thousand heads, accounting for 3.7% of the herd (Figure 2).

### 3.2. Production System and Scale

Goats in Vietnam are raised under three different production systems, namely: extensive, semi-intensive, and intensive [12,13,33]. Extensive farming is a traditional system in which indigenous goats are raised for meat. Goats seek and select feed for themselves independently in native pastures, forests, and communal grazing areas. There is a low-input investment in terms of breeding, stables, feed, veterinary service, and labour, an outstanding advantage of this farming system. Goats raised extensively tend to be highly preferred and relatively high-priced in domestic markets. However, they have low growth and reproductive performance, so their economic efficiency is not high [33].

The semi-intensive system is the most popular, with approximately 70% of goat farmers using this method [12]. In this system, goats are extensively grazed during the daytime and supplemented with additional feed in stables at night time. This system is suitable for smallholder dual-purpose goat production in most of the country’s coastal and midland regions [33]. Nguyen et al. (2021) [13] reported that the system can effectively use the locally available feed resources and crop by-products, but it requires higher costs in terms of labour and stables in comparison to the extensive production system.

In the intensive farming system, goats are kept in their stables and surrounding yards most of the time and their feed is supplied by producers. Raising goats under this system offers benefits such as no grazing areas required, higher biosecurity, and a lower risk of physical injury compared to the other systems [33]. Nevertheless, high levels of investment, as well as husbandry and management skills, are the main barriers to developing this production system.

Smallholder farming is the most prevalent level of goat production. Goats are raised by more than 417,000 households across the country (Figure 3). There were about 306,000 of these households raising fewer than 10 goats each, accounting for 73.4%. Some 97 thousand households raised between 10 and 29 heads, equivalent to around 23.3%, while 2.6% of households raised between 30 and 49 heads. The number of households raising more than 50 heads/household accounted for only 0.8% [16]. In recent years, a number of large farms, with between 1000 and 3000 meat goats each, have appeared in Lam Dong, Ninh Binh, and Long An [16]. Nguyen et al. (2021) [13] stated that there has been a transition, from extensive smallholder goat keepers to intensive medium- or large-scale producers. The Vietnamese government planned to expand both the goat and sheep populations by about 4.5 million heads in 2030, 90% of whom would be from crossed breeds and would be mainly raised in medium- and large-scale farms [24].

### 3.3. Main Goat Breeds

Co and Bach Thao constitute the two primary indigenous goat breeds in Vietnam [17,34]. The Co breed, which is found nationwide, is predominantly reared for meat production. In 2018, the Co goat herd consisted of more than 0.8 million heads and accounted for about a third of the total goat population [35]. Co goats have relatively good reproductive performance (1.7–2.5 kids/female/year) but are limited by their small body size (30–40 kg for males and 25–35 kg for females), low milk production (45–60 litres/cycle), and low lean meat percentage (31.7%) [36,37]. They adapt well to natural environments and outdoor conditions and can eat a wide range of grasses and leaves. Their advantages include good tolerance and adaptability to many different environments and good disease resistance. Therefore, the Co goat breed has been widely raised extensively or semi-intensively in remote, mountainous, and ecologically vulnerable regions.

The Bach Thao breed is considered a dual-purpose breed, intended for both meat and milk production. It has a smaller population size than the Co breed, approximately 60 thousand heads [17]. Traditionally, they have been raised mainly in the Ninh Thuan and Binh Thuan provinces. Over the last decade, the breed has been developed in many provinces and cities throughout the country [35]. In comparison with Co goats, they have better fertility (2.5–4.0 kids/female/year), a larger body size (55–75 kg for males and 40–50 kg for females), higher milk production (150–320 litres/cycle), and higher lean meat percentage (33–35%) [36]. They are also able to adapt to a wide array of environmental conditions and farming methods [35,36]. Recently, the Bach Thao has been crossed with exotic meat-type breeds, such as Boer and Kiko, for meat-oriented offspring, or with exotic milk-type breeds, like Saanen and Alpine, in order to improve the milk yield of their progenies [35].

High-performance exotic goat breeds have been introduced by the government through national livestock breeding improvement programs to enhance the body size and productivity of indigenous animals. Exotic breeds, like Boer, Kiko, Saanen, Alpine, Barbari, and Jamunapari, were imported with the purpose of enhancing meat and milk productivity through crossbreeding [36,37]. A number of studies agreed that genetics is one of the most important factors affecting an animal’s growth performance, carcass traits, and meat quality [37,38,39]. Both Mui et al. [40] and Hung et al. [37] proved that exotic crossbred goats have a higher body size, dressing, and lean meat percentages than pure Co goats. Moreover, Hai and Binh, 2019 [41] reported that the growth rates of Boer and F1 (Boer × Bach Thao) goats are higher than that of pure Bach Thao goats. While these goat breeding programs have contributed to the improvement of crossbred offspring from local goats and partially addressed the demand for goat meat and milk; there is a concern about the indigenous breeds being genetically diluted as a result of indiscriminate crossbreeding [17].

## 4. Market Status and Types of Goat Supply Chains

### 4.1. Market Status

In recent years, the number of slaughtered goats in Vietnam has steadily increased by 29.1%, from 1.10 million heads in 2017 to 1.42 million heads in 2021, despite the substantial influence of the COVID-19 pandemic [32]. The yearly supply of goat meat has witnessed a steady growth, rising from 18.1 thousand tons in 2014 to 35.3 thousand tons in 2021, equivalent to an annual increase of 13.6%. On average, the goat meat yield supplied to the market has increased by approximately 2.5 thousand tons annually. This growth could be primarily driven by increased domestic demand, due to a 47% surge in GDP per capita [42] and a 7% population growth [43] from 2014 to 2021.

As demand increases, goat traders are forced to import [21]. According to Trend Economy (2021) [44], Vietnam has annually spent about USD 4–8 million importing sheep and goat meat through formal market channels from Australia, Mongolia, New Zealand, Hong Kong, and Kenya in the last decade. Within these formal channels, most of the products are frozen and chilled meat from Mongolia and Australia. The United Nations Comtrade Database (UNCD) [45] reported that Mongolia was the largest supplier of frozen and chilled goat meat, with around USD 1.6 million worth of goat meat exported in 2020, while Vietnam spent more than USD 0.3 million purchasing frozen and chilled goat meat from Australia in the same year.

There has been an increase in imported live goats, mainly from Laos, to address the high demand for freshly sourced goat meat in Vietnam [23,46]. In 2014, it is estimated that only about 500 live goats were imported, while the number increased to more than 36 thousand goats in 2017. Currently, imported goats are estimated to total approximately 40 thousand heads per year [16]. According to Gray et al. (2019) [47], a significant proportion of goats produced in certain survey areas of Laos, as high as 90%, are transported to Vietnam, with their average price being 30% higher than that of Vietnamese crossbred goats. Most of these transactions are unreported and do not go through commercial border gates [16].

Additionally, the goat milk yield has not met the domestic market demand [48], even though goat milk production was estimated to increase from 389 tons in 2006 to 1551 tons in 2015 [49]. The demand is surging, as many consumers believe that goat milk has medicinal and health properties, so goat milk retails at a price range of 1.5 to 3 times higher than cow milk [48,50]. Consequently, goat milk production is expected to witness an ongoing growth in line with the development of the national economy in the coming years [23,50].

It is worth noting that Vietnam’s goat industry is primarily focused on meeting domestic demand for meat and other goat products. While goat milk is mainly supplied to metropolitan areas, such as Hanoi and Ho Chi Minh City, goat meat is consumed widely in restaurants and household feasts across Vietnam. Domestic customers prefer meat from native goat breeds to others, as they believe that native goats raised extensively or semi-intensively are cleaner, healthier, tastier, and have a higher nutritive value than goats from exotic breeds or those which are raised intensively [19]. When analysing the chemical compositions of 12 striploin samples from each goat breed raised in an extensive production system, Hung et al. (2014) [37] found that Co goats had significantly higher crude protein and essential amino acid contents than Boer crossbred goats. Goat meat is also used in traditional cuisine for special occasions, such as weddings, festivals, family reunions, and religious ceremonies throughout the year [33].

Even though the industry is mainly focused on meeting domestic demand, Vietnam also exports a small volume of goat meat. According to the UNCD (2023) [51], Vietnam recently exported goat meat to Japan, India, Greece, China, and the United Kingdom.

### 4.2. The Types of Goat Supply Chains

In Vietnam, goats are traded and processed through both formal and informal market channels, either directly from producers to consumers or with the involvement of intermediaries. The formal market is subject to regulation by multiple governmental bodies at both the local and national levels, such as the Department of Livestock Production (DLP), the Department of Animal Health (DAH), Provincial Departments of Agriculture and Rural Development, and Local Market Management Boards. The DLP, a subdivision of the Ministry of Agriculture and Rural Development (MARD), holds the authority to issue trading permits for livestock in accordance with relevant legislation. The Department also administers quota schemes for importing and exporting livestock and their products. However, the official records on the trading data and marketing schemes of the goat industry have received limited attention. 

In contrast, goat marketing is mostly informal and dominated by small-scale farmers and traders. Within the informal market, trading occurs on a need basis, with individuals primarily selling goats for emergency income or to open market traders involved in processing and selling goat meat. The flows of live goats and processed goat products are quick, as consumers prefer fresh chevon slaughtered within a day [27]. Prices are determined through negotiations between the buyer and the seller, resulting in significant variations [52]. Trading in informal channels may avoid food safety inspection, taxes, and institutional costs. Consequently, the organised trading of goats, as well as related marketing structures and infrastructure, is not well established, resulting in limited availability of detailed information regarding the market chain stakeholders, the prevailing market prices, transaction costs, and the specific flow of goats and goat products. Van and Don (2021) [53] stated that it becomes challenging to ascertain the quantity of goats being traded, due to the absence of recorded marketing information. In some cases, farmers have become vulnerable to buyers who offer significantly low prices for their livestock. 

## 5. Functions of Stakeholders Involved in Goat Supply Chains

In the agricultural industry, the structures of goat supply chains are considered open networks [54], which means that there is only a one-directional physical flow of the product from producer to consumer through a number of direct stakeholders [55], such as input providers, producers, traders, processors, wholesalers, and retailers, who implement a number of functions, including input supply, production, trading, processing, and distribution. Furthermore, in Vietnam, there are groups of supporting entities for the goat supply chain, including transporters, policy framework supporters, and research and technical supporters. The goat supply chain is summarised in Figure 4. However, the way in which these stakeholders connect and interact with each other can vary significantly [54].

### 5.1. Main Functions of Stakeholders

#### 5.1.1. Input Supply

The input demand for the smallholder goat farms is small, as they require less purchased feed, and most of the goat feed is provided by their own farms [27]. Therefore, supplementary feed and veterinary medicines are involved in the supply of other livestock, such as pigs, poultry, and cattle. The majority of suppliers use leased locations for their business activities, and their merchandise is sourced from animal feed and veterinary pharmaceutical companies [56]. The input is procured and sold to customers through cash transactions, with deliveries made to local buyers including transportation costs incorporated into the input’s overall cost. When selling to customers, they are also advised on how to use and preserve the product.

#### 5.1.2. Production

As mentioned previously, the goat industry is mostly based on small-scale production, with the vast majority of farmers raising fewer than ten goats in semi-intensive systems. Most of them are independent farmers who raise goats without any contract with traders and/or abattoir owners or restaurateurs [16]. Therefore, they can freely sell their goats to any stakeholders [57], but there was not much information available regarding the role or power of farmers in price negotiations.

#### 5.1.3. Trading

Live goat traders are the main buyers of goats from smallholder farmers in surrounding areas or adjacent provinces. The majority of them are small-scale, with an amount of working capital from VND 10 to 30 million (equivalent to USD 400 to 1200) to serve the trade [57]. Once they have reached the targeted number of animals, the goats are then transported on motorbikes and sold to abattoirs and/or other traders (Figure 4). There are also traders who import goats and goat products and sell them to supermarkets, goat meat outlets, and restaurants. Most goat traders work independently, while some of them depend on the abattoirs [20].

#### 5.1.4. Slaughtering/Processing

It seems that there is no clear evidence or reporting on the goat processing industry. The processes of goat slaughtering and preliminary processing are conducted through abattoirs. They can be restaurant-based, butcher-based, or professional. Restaurant-based and butcher-based abattoirs are usually small-scale slaughterhouses, which only slaughter fewer than ten goats per day and serve one or several restaurants and butchers. It is believed that abattoir owners have the most powerful influence over goat prices along the supply chain [47]. The abattoirs purchase goats from the traders and/or directly from farms, and they sell the whole carcasses or meat cuts to restaurants, wet traditional markets, and supermarkets (Figure 4). The prices proposed by them to the traders, and consequently passed on to the farmers, will align with the market equilibrium for a specific day [20]. 

#### 5.1.5. Distribution 

There is a wide network of goat distributors in Vietnam, with a lot of restaurants, wet traditional markets, supermarkets, and goat meat outlets involved [20]. The distribution flows after slaughter are quick, as consumers traditionally prefer fresh goat products [27]. However, commercial slaughtering guidance and official quality standards for goat products have received limited attention in Vietnam. This has led to difficulty controlling and inspecting the food safety and quality of goat meat and milk [20,48].

### 5.2. Supporting Stakeholders of the Supply Chain

Research and technical supporters involve research projects on and the techniques of livestock management, breeding, and feed preparation, which help farmers to tackle specific challenges related to disease control, protect livestock from adverse conditions, and mitigate the impact of natural disasters [58]. They also provide recommendations to promote and strengthen livestock value chains in general, the goat supply chain in particular, in Vietnam, and to improve the capacity for local researchers and goat producers through training courses/workshops. Such supporters include agricultural research institutions, such as the National Institute of Veterinary Research (NIVR), the National Institute of Animal Science (NIAS), and the National Agricultural Extension Centre (NAEC); local government extension offices; universities, such as the Vietnam National University of Agriculture (VNUA); veterinarians; and non-government organisations (NGOs), including the Australian Centre for International Agricultural Research (ACIAR), the Asian Development Bank (ADB), the International Livestock Research Institute (ILRI), and the International Centre for Tropical Agriculture (CIAT).

Stakeholders of the livestock chain in general, and of the goat supply chain in particular, can access credit from a number of financial organisations or associations in order to establish and expand their businesses [59], although the goat industry has not received as much investment interest as the pig, dairy cattle, and poultry industries. Credit can come from state or commercial banks, such as the Bank for Social Policies or Agribank; microfinance institutions; people’s credit funds; NGOs; and other credit funds and organisations [59]. Accessing capital from credit institutions with suitable interest rates can be difficult [60], so many stakeholders who are unable to approach formal capital institutions have to obtain access through informal credit sources, such as black credit with high-interest rates. This approach can become a risk that hinders the sustainable and effective development of the business.

Transportation plays a crucial marketing role along the goat supply chain; primary activities involve the transportation of feed, live animals, goat carcasses, and other goat products on behalf of buyers, in addition to them being traders as well. The three major means of transportation are trucks (with which input suppliers and traders transport feed and farming supplies), coaches (with which abattoirs transport goat carcasses and other edible tissues), and motorbikes (with which small-scale traders transport feed and live goats locally). Their customers consist of feed suppliers, goat producers, restaurateurs, and abattoir owners, with transport fees calculated per load rather than per individual goat. According to Circular 101/2020/TT-BTC, ruminants are officially mandated to obtain movement permits, with the current cost of these permits estimated to be around VND 50,000 (USD 2)/lot to be transported. However, it seems that no official standards and guidelines for goat transport have been established, and animal welfare during transportation has received limited attention. 

In Vietnam, policy and quality standard supporters are government agencies that regulate the goat industry. At the national level, they include the National Assembly, MARD divisions (DLP, DAH), and the Vietnam Food Safety Authority (VFA). At the local levels, there are the Provincial People’s Councils, Provincial People’s Committees, Provincial Departments of Agriculture and Rural Development (DARDs), Sub-Departments of Animal Husbandry and Veterinary Medicine, and District Sub-Departments of Agriculture and Rural Development. Long et al. (2020) [61] stated that these agencies create policy environments and quality standards through decrees, decisions, directives, procedures, resolutions, etc., and establish a development plan aimed at enhancing and supporting livestock production as a whole, with a specific focus on goat production.

Analysing goat supply chains helps to identify different stakeholders and channels that show the flow of goat products, as well as the constraints faced by each stakeholder [62]. Figure 4 shows that the goat supply chain is complex, including many direct and supporting stakeholders. However, there is neither close linkage nor significant input spent by stakeholders toward helping to develop the goat industry. There is also not much information or documentation regarding either horizontal or vertical links within the supply chain. There is also not much evidence of linkage among the direct actors, such as between farmers and traders, or with traders among restaurateurs and outlets [20]. Smallholder goat farmers face significant challenges when attempting to participate in new or established supply chains. Despite high demand, farmers must acquire the necessary skills for production, business management, and entrepreneurship in order to elevate their operations to an enterprise level and effectively engage in and benefit from goat value chain development. Gray et al. (2019) [47] reported that goats already provide a steady source of income, and when fully integrated into the value chain, farmers have the potential to enhance their livelihoods and establish sustainable enterprises.

## 6. Challenges to Goat Production and Marketing

### 6.1. Challenges to Goat Production

There is a lack of sustainable breeding programs in Vietnam. A significant proportion of households have been relying on a single male for breeding purposes in their herds for an extended period [19]. Inbreeding and uncontrolled mating are rampant in communal areas. Kosgey et al. (2006) [63] stated that the efficacy of small ruminant breeding methods within smallholder production systems is limited due to factors such as the utilisation of single-sire flocks, insufficient animal pedigree and performance recording, the absence of systematic animal identification, and organisational deficiencies. In addition, the infrastructure and projects crucial to gathering dependable performance and pedigree data are scarce and often unavailable. Indigenous breeds have small body sizes, poor growth performance, and low meat yield compared to exotic breeds, which, on the other hand, tend to exhibit a deficiency in the adaptive characteristics required for survival and productive purposes. It is crucial to contemplate well-designed schemes for goat genetic improvement that effectively address these challenges [64].

The development of ruminant production is also confronted with a shortage of forages. Nguyen et al. (2020) [65] stated that, in the dry or winter season, pastures and cut grasses only meet about 35–57% of the total forage demand. Moreover, the quality and availability of forages are neither uniform nor favourable all year round. As a result, ruminant diets need to be supplemented with concentrate in times of forage shortage in order to maintain their performance.

The producers’ awareness of biosecurity and proactive disease prevention and control for goats is still limited, due to a high proportion of small-scale extensive production systems [27]. There is a wide range of causative agents, even if preventive health care facilities for goats are inadequate in Vietnam. Diseases including diarrhoea, parasites, congestion, bloating, goat pox, and foot and mouth disease are prevalent and regarded as constant and considerable issues in the goat production system [66]. Limited veterinary and extension services are additional constraints, especially in remote and mountainous areas. Although state veterinarians and extension officers are available, most goat producers rarely contact them unless they suspect a notifiable disease. Consequently, the prevalence of infectious diseases and lack of veterinary services constitute major challenges to goat production in Vietnam.

### 6.2. Challenges to the Goat Market

In Vietnam, the goat market seems to receive relatively little attention in comparison to the industry’s economic contribution [16]. In the informal sector, goats seem to be traded conveniently, and related actors can avoid taxes, institutional costs, and paperwork procedures, which are time-consuming. This renders formal markets unappealing for domestic goat sales. Consequently, the development of formal goat markets is deficient and lagging and, in certain instances, absent [53]. This situation discourages motivation and lacks the incentives to enhance the productivity and market infrastructure of the goat industry.

The limited quantity of goats owned by individual farmers, particularly in remote mountainous regions, presents a significant constraint as it hinders the assurance of a consistent supply. Moreover, producers in these areas do not prioritise market-oriented production as they only sell their livestock when they require immediate cash [27]. This situation makes it challenging for middlemen to operate in these areas as the quantity supplied is contingent upon sporadic demands for cash among the producers. On the other hand, the output price of livestock products fluctuates and is unstable, causing many livestock farmers to reduce the size of their herds or even stop raising.

There is a scarcity of market information relevant to prices and marketing channels, which poses challenges for small-scale goat farmers in selling their animals within formal markets [67]. Additionally, the absence of marketing infrastructure for goats, including auction pens, results in the current practice of marketing goats in open-air markets that are familiar to buyers. Research on goat marketing and the value chain in Vietnam remains limited.

One of the challenges for ruminant markets in general, and the goat market in particular, is inadequate and mainly small-scale slaughtering systems. Nga et al. (2017) [68] reported that only 6.4% of the 1882 ruminant slaughtering locations across the country slaughter and process more than 10 heads/night, so the processing capacity is still low. Furthermore, there are no official goat slaughter protocols or meat quality standards. Such circumstances could potentially raise concerns among consumers regarding food safety, as well as meat quality and traceability [69]. The sustainability of small-scale slaughterhouses could be at risk, as they may encounter challenges in terms of the necessity to upgrade their facilities and modify their practices and attitudes regarding animal slaughtering and meat handling [61,68].

Goat milk and its derivatives are rarely found in formal marketplaces, such as well-established convenience stores and supermarkets, in Vietnam, even though they have traditionally been consumed by rural populations in many other developing countries. In the last decades of the 20th century, the dairy goat market developed very well in the whole country, mainly for local consumption [23,26]. However, when imported milk sources became abundant and dairy cow production developed, producers moved toward a tendency to produce more meat goats [23]. Thu (2017) [26] stated that goat milk is mainly consumed fresh or used to make cheese and yoghurt, but its yield has not been included in official statistics.

By integrating lower tariffs, non-tariff market access, and standardised trade regulations, free trade agreements, such as the European Union–Vietnam Free Trade Agreement (EVFTA) and the Comprehensive and Progressive Agreement for Trans-Pacific Partnership (CPTPP), created new opportunities for goat-exporting countries, including Australia, Canada, and New Zealand [70]. These countries have a significant advantage in animal productivity, production cost, and prices, and they can become the main suppliers of chevon and other goat products in the future. However, the free trade agreements are a major challenge to domestic smallholder goat producers in Vietnam. 

## 7. Opportunities for the Development of Goat Production and Marketing

The rise in global and domestic demand for goat meat can be attributed to population growth and increasing income levels [23,71]. As consumers prioritise their health, there is a growing demand for chevon and other goat products, owing to their nutritional advantages and values. According to Mazhangara et al. (2019) [71] and Mariuset al. (2021) [52], chevon, goat milk, and goat cheese are characterised by health benefits, such as lower levels of total fat, saturated fatty acids, and cholesterol, as well as higher levels of polyunsaturated fatty acids which make them more nutritional, allowing them to compete with other higher-priced red meat and ruminant products. Therefore, the industry is experiencing considerable growth and expansion as supplies have not met the higher demand imposed by consumers [52,72].

The adaptability of goats to the harsh environment, and their broad-spectrum feeding habits, can enhance livestock producers’ resilience to the situation of forage shortages in the winter or dry season, as well as their resilience to agricultural land pressure, which is increasing in Vietnam [65]. Darcan and Silanikove (2018) [73] and Giao (2018) [19] reported that goats tend to cope better with a shortage of forages than do cattle and buffaloes, due to their typical feeding behaviour and inherent capability to eat a wide range of forages.

A number of studies agreed that goats possess inherent physiological adaptations that enable them to thrive in challenging environments where crop cultivation and other forms of livestock production may be challenging or unsuitable [71,72,73]. Apart from grasses and legumes, goats have the ability to consume and digest tannin-rich browse plants efficiently and effectively extract essential nutrients from them [74]. They also exhibit an ability to withstand extended periods without access to water, demonstrating their resilience in water-deprived conditions and very high temperatures. Due to their resilience and positive attributes, goats are favourable for smallholder producers with low input costs, especially those inhabiting resource-poor regions that are semi-arid or arid and typically associated with limited agricultural possibilities [52,67].

There has been a transition from extensive to cut-and-carry in intensive goat farming systems [13,19]. In the latter systems, goats often receive more balanced diets and supplements, which help enable the animals to meet their protein, energy, and mineral requirements. This could increase the capacity of the goat population and its genetic potential, with less dependence on the communal pastures and grazing land that are coming under increased demand in Vietnam.

Vietnam’s current agricultural restructuring policies regarding livestock development encompass two approaches: (i) promoting large-scale and intensive production in specialised regions utilising advanced technologies, and (ii) supporting household livestock production while encouraging the adoption of technologies and enhancing biosecurity measures [47]. The Vietnamese government strongly supports the development of goat production and plans to increase the total goat population, especially in regions with high potential. The government aims to consolidate the industry by fostering the establishment of larger farms, with the goal of ensuring a stable supply of livestock with stable pricing. Additionally, the government plans to offer farmers insurance coverage for their flocks, providing them with a measure of financial protection. This would also encourage investments from the private sector. With the intent to protect goats from forage shortages and climate change, animal scientists, veterinarians, and extension officers are actively seeking alternative food sources and improving shelter for goats. Moreover, the Vietnamese government is supporting farmers by facilitating the importation of resilient goat breeds through a number of government decrees and decisions on animal breeding and development strategy, as well as through the promulgation of specified high-yield livestock breeds [13].

As previously mentioned, the appearance of goat meat in large distribution channels such as supermarkets and convenience stores is limited in Vietnam. Moreover, Vietnamese goat producers and traders can access vast international markets with reduced or waived taxes and tariffs, thanks to the effects of free trade agreements [70]. By exploiting these channels and markets wisely, there is huge potential for producers to help increase goat meat consumption.

## 8. Conclusions and Recommendations

### 8.1. Conclusions

Goat production in Vietnam is mainly small-scale, although there have been substantial increases in the total national goat population, as well as in the consumption and on-farm pricing of goat products. The goat market is driven by domestic demand, but the accurate determination of the size of the goat market is difficult due to a shortage of official market records and research. Goat marketing systems are poorly developed because these activities are mostly informal and dominated by small-scale farmers and traders. The goat industry is dealing with many challenges, including forage deficiency, inconsistency in supply, limited market infrastructure and necessary studies on it, a lack of price incentives and suitable processing systems, and competition from foreign suppliers. Nevertheless, a number of opportunities and a great amount of potential exist for improving goat production, such as goats’ adaptability and ability to thrive in harsh environments, increasing domestic and international demands, and strong support policies from the government. Increasing health consciousness amongst meat consumers and improvement of the market infrastructure, as well as the lower input cost and higher value addition compared to other livestock, are also factors that have the potential to bolster the continual development of goat production and marketing in Vietnam.

### 8.2. Recommendations

Suitable genetic improvement strategies for indigenous goats are crucial in order to enable producers to not only gradually meet the increasing market demand, but also keep satisfying domestic consumer preferences. There is a need for clarity on how the indigenous goat phenotypes or management types can result in premium products. This information is important for enabling producers to make informed decisions on whether to focus on preserving the indigenous goat phenotypes or improve their goats’ genetic potential through cross-breeding. To enhance the breeds for better adaptability and productivity, artificial insemination and introducing breeding bucks that have desired traits could be pragmatic solutions.

Capacity improvement for the goat supply chain actors is necessary, with training programs focusing on: (1) better feed supplementation, herd management, and disease control for farmers; (2) improving slaughtering and meat handling practices, which involves implementing credible meat inspection protocols and enforcing strict compliance with hygienic standards for abattoirs, restaurants, and other goat processors; and (3) the ways to approach and exploit market information and credit resources for the main actors of the supply chain. It is also necessary to improve awareness about the adaptability of goats to climate change and the resilience of the stakeholders to animal and human pandemics, as well as natural disasters.

Research and collaboration amongst supply chain stakeholders, especially strengthened links among animal scientists, extension officers, policymakers, traders, and producers, are needed in order to accurately forecast current and future demand for goat products and to help prevent an oversupply crisis that would lead to a price collapse. The involvement of both enterprises and cooperatives in the goat value chain is required in order to build brands for goat products in each region and ensure stable output for goat farmers.

The government must invest in research on the livestock sector, such as feeding, breeding, management, and veterinary services, as well as goat market assessment. The government should also provide subsidies to vulnerable actors in the event of disease outbreaks, price collapses, and natural disasters.

## Figures and Tables

**Figure 1 animals-13-02546-f001:**
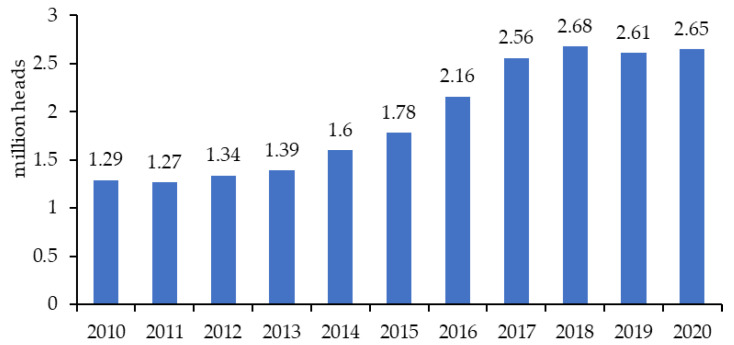
The total goat population in Vietnam from 2010 to 2020 (adapted from Nguyen (2022) [7]).

**Figure 2 animals-13-02546-f002:**
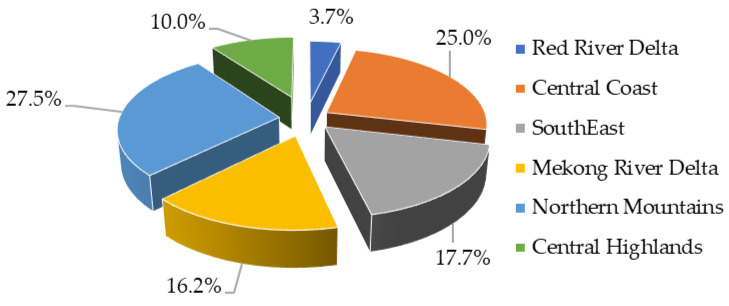
The regional distribution of goats in 2021 (source: GSO (2022) [32]).

**Figure 3 animals-13-02546-f003:**
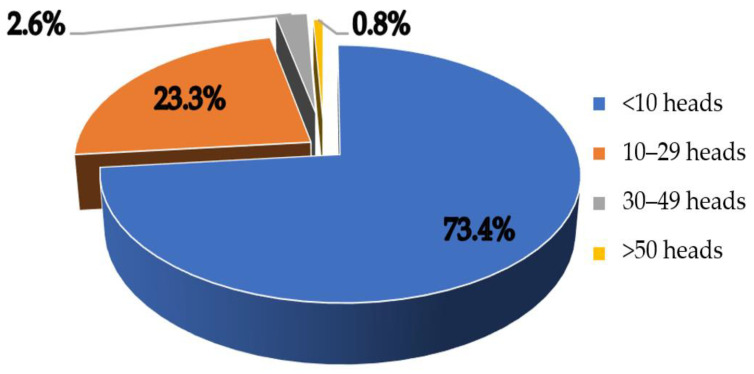
Goat raising scale in Vietnam in 2020 (adapted from Khoi (2021) [16]).

**Figure 4 animals-13-02546-f004:**
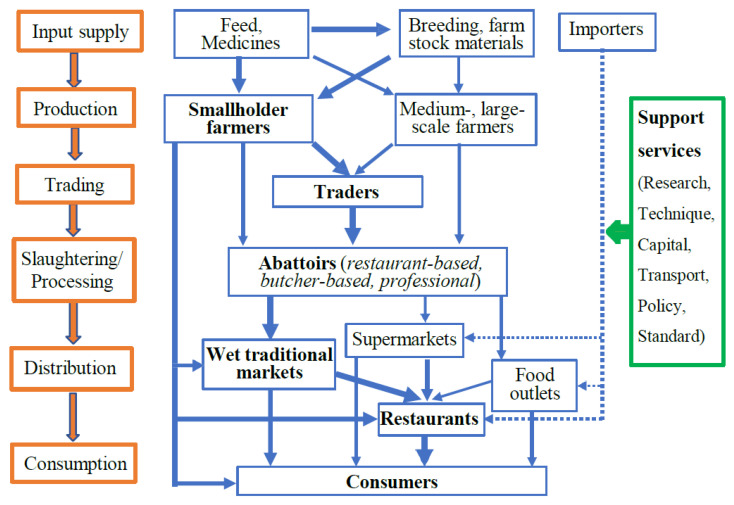
Goat supply chain map for Vietnam. The bolds and thicker arrows indicated the main stakeholders and the main follow of the goat supply chain.

## Data Availability

Not applicable.

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
