# Peer review of "Goat Production, Supply Chains, Challenges, and Opportunities for Development in Vietnam: A Review"

_animals, 2023, doi:10.3390/ani13152546_

Round 1
Reviewer 1 Report
This paper provides valuable information on an important topic “goats sector in Vietnam”. This will have important benefits for researchers in the field.
I have some comments that I hope the authors respond to:
Introduction:
This section should be rewritten because the authors used paragraphs from some published papers and they change only data.
Data about goats in Asia and neighboring countries should be given.
The authors should more detailed about the goats sector in Vietnam.
Methodology:
This section should be more detailed. The systematic review process should be presented with different steps according to the PRISMA guideline.
Results and Discussion
Goat production systems in Vietnam should be detailed and presented in a separate sub-section
What about diseases and parasites?
Author Response
Dear Reviewer #1,
The authors really appreciate your precious time in reviewing the manuscript and providing valuable feedback. We tried our best to address every comment. Please see the attachment.

Reviewer 2 Report
It is an expert report, that is, the presentation of statistical data.
The paper has no characteristics of scientific work or scientific contribution.
Author Response
Dear Reviewer #2,
The authors really appreciate your precious time in reviewing the manuscript. The Reviewer is off-course entitled to his/her personal opinion, but we believe we have provided enough empirical evidence-based scientific data justifying that the manuscript is a literature review paper. We tried our best to address the comment. Please see the attachment.

Reviewer 3 Report
I found this article interesting and useful. However, the authors should work more on the structuring of the article to avoid repetion the same statements

Please find comments attached
Author Response
Dear Reviewer #3,
The authors really appreciate your precious time in reviewing the manuscript and providing valuable feedback. We tried our best to address every comment. Please see the attachment.

Round 2
Reviewer 1 Report
All the comments have been addressed.
Author Response
Authors are glad that our responses satisfactorily addressed the Reviewer’s comments and suggestions. Once again, we are thankful to your valuable time and feedback.
Reviewer 2 Report
This is a manuscript in the category of review articles. The quality of such articles depends on the applied methods of analysis of available publications and the quality of these publications. In this case, the up-to-date methodology was not applied, the research and publications to which the manuscript refers are mostly not based on scientific research. Therefore, the presented results and conclusions should represent professional assessments only and should be interpreted in that way. The paper, therefore, represents a contribution to the systematization of information on goat farming and the supply chain of goat products in Vietnam, but it does not have an essential scientific contribution.
Author Response
Authors are grateful to the reviewer for his/her precious time and comment. We completely agree that the quality of a review manuscript depends on the quality of previously available publications and the applied analysis methods. In this manuscript, we used an inductive thematic analysis method, which was created by Braun and Clarke (2006) and have been used widely in reviewing literature, to identify, analyse, and discuss the common themes, opportunities and challenges of the Vietnamese goat industry. We also have tried our best to search and collect statistical reports and scientific publications, and provide the empirical evidence-based scientific data. In scientific perspective, the review manuscript might be less comprehensive than what the review expected. This could be because of the lack of goat research and statistical data, especially in marketing and value chain as mentioned in the manuscript. However, the paper can provide the information base for further research on the goat-related issues in Vietnam.
Reviewer 3 Report
I have read the revised manuscript, which has substantially improved. Still, in my view, the authors should work more on it. It is not yet ready for publication. Authors should edit the article and make it more fluent. Besides, there are still too many repetitions, and some information is inconsistent, such as the description of breeds. Meat from local goats fetches higher prices, and they are hardy. Still, authors argue that exotic breeds must replace goats because productivity is higher per animal. But heavier animals also need more feed; hence, I would prefer a broader discussion. What about discussing productivity per 100 kg BW? Several more miniature goats also reduce the risk compared to heavier exotic ones. Also, they need less purchased feed (line 306). Line 574 is fine; make it consistent. The authors should streamline information about the market and value chains and reduce repetitions. The goat number has increased significantly (13.6%), but the authors complain. Line 329 is unclear; please explain. Line 257, different meat quality - how solid was this experiment?
I have read the revised manuscript, which has substantially improved. Still, in my view, the authors should work more on it. It is not yet ready for publication. Authors should edit the article and make it more fluent. Besides, there are still too many repetitions, and some information is inconsistent, such as the description of breeds. Meat from local goats fetches higher prices, and they are hardy. Still, authors argue that exotic breeds must replace goats because productivity is higher per animal. But heavier animals also need more feed; hence, I would prefer a broader discussion. What about discussing productivity per 100 kg BW? Several more miniature goats also reduce the risk compared to heavier exotic ones. Also, they need less purchased feed (line 306). Line 574 is fine; make it consistent. The authors should streamline information about the market and value chains and reduce repetitions. The goat number has increased significantly (13.6%), but the authors complain. Line 329 is unclear; please explain. Line 257, different meat quality - how solid was this experiment?
Author Response
Authors are thankful to the reviewer’s comprehensive, invaluable comments. We are responding to his/her comments in detail in the attached file.
